# Burden, and trends of breast cancer along with attributable risk factors in Gulf Cooperation Council countries from 1990 to 2019 and its projections

Majed M. Ramadan[1], Heba A. Alkhatabi[2], Doaa Aboalola[3,4], Samkari Alaa[5], Rawiah A. Alsiary[3,4]*

1 Population Health Research Section, King Abdullah International Medical Research Center, King Saud Bin Abdulaziz University for Health Sciences, Jeddah, Saudi Arabia, 2 Faculty of Applied Medical science, King Abdul Aziz University. Hematology Research Unit (HRU), King Fahad Medical Research Center (KFMRC), 3 Department of Cellular Therapy and Cancer Research, King Abdullah International Medical Research Center, Jeddah, Saudi Arabia, 4 King Saud Bin Abdulaziz University for Health Sciences, Jeddah, Saudi Arabia, 5 Department of Pathology and Laboratory Medicine, King Saud Bin Abdulaziz University for Health Science, King Abdulaziz Medical City, Ministry of National Guard- Health Affairs, Jeddah, Kingdom of Saudi Arabia

* alsiaryr@kaimrc.edu.sa

## Abstract

### Introduction

Breast cancer (BC) is a growing global public health concern, affecting millions of women worldwide. Gulf Cooperation Council (GCC) countries are no exception to this trend. Mortality rates in GCC nations are still high despite improvements in BC treatment. This article examines the changing picture of BC incidence, prevalence, and mortality in the GCC region from 1990 to 2019 and predictions up to 2030.

### Material and method

Using data from the Global Burden of Disease study, we analyzed BC incidence, prevalence, and mortality rates per 100,000 individuals across different age groups and countries.

### Results

The study reveals a significant rise in Age-Standardized Incidence Rates (ASIR) for breast cancer among females in Saudi Arabia from 1990 to 2019, with Oman experiencing the highest increase and Kuwait the highest decrease. Bahrain also saw a significant increase in male Age-standardized death rate (ASDR), despite all other countries experiencing a decrease. Also, the data demonstrated a statistically significant positive correlation between ASIR and Human Development Index (HDI), evident across all countries. Metabolic risk and tobacco use were identified as primary

**Data availability statement:** The Global Burden of Disease (GBD) 2019 database, available through the Global Health Data Exchange (GHDx) at https://ghdx.healthdata.org/gbd-2019, supplied the data used in this research. The dataset includes incidence, prevalence, mortality, and disability-adjusted life years (DALYs) for breast cancer in the Gulf Cooperation Council (GCC) countries, as well as associated risk factors such as dietary risks, low physical activity, metabolic risks, and tobacco consumption, which form the essential dataset underpinning the conclusions of this paper. No special access permissions were required, and this data is publicly available. The Supporting Information file (S1 Table) includes all the data utilized to generate the figures and tables presented in this publication, encompassing prevalence, age-standardized death rates (ASDR), age-standardized incidence rates (ASIR), and DALYs categorized by year and sex. By directly obtaining the raw GBD 2019 data from GHDx and applying the statistical methods described in the Methods section, researchers who are interested can fully replicate our studies.

**Funding:** The author(s) received no specific funding for this work.

**Competing interests:** The authors have declared that no competing interests exist.

**Abbreviations:** DR, Death rate; IR, Incidence rate; ASDR, Age-standardized death rate; ASIR, Age-standardized incidence rate; ASR, Age-standardized rate; BAPC, Bayesian age-period-cohort; BAMP, Bayesian age-period-cohort modeling and prediction; BC, Breast cancer; BMI, Body mass index; CI, Confidence interval; DALYs, Disability-adjusted life-years; EAPC, Estimated annual percentage change; FPG, Fasting plasma glucose; GBD, Global Burden of Disease; HDI, human development index; UI, Uncertainty interval.

contributors. A ten-year BC prediction predicts a significant increase in female cases, with Saudi Arabia expected to experience the highest rise.

## Conclusion

This study underscores the urgent need for improved BC awareness, early detection through screening programs, enhanced access to quality healthcare services, and the addressing of sociocultural barriers in the GCC countries.

## Introduction

Breast cancer is one of the foremost public health concerns globally, affecting millions of women worldwide [1]. Gulf Cooperation Council (GCC) countries, comprising Saudi Arabia, Kuwait, the United Arab Emirates, Qatar, Bahrain, and Oman, are no exception to the rising prevalence of breast cancer [2]. The GCC countries have their unique demographic, economic, and healthcare contexts, which differ significantly from other Middle Eastern countries. The GCC countries have experienced rapid economic growth, urbanization, and lifestyle changes, leading to distinct epidemiological transitions and healthcare challenges, particularly in cancer burden. Despite considerable advancements in breast cancer management, mortality rates remain a significant concern in GCC countries. Research indicates that breast cancer incidence rates in GCC countries have been increasing steadily over the past few decades. This rise in incidence is associated with several factors, including changing lifestyle patterns, increased adoption of Westernized diets, evolving reproductive behaviors, urbanization, late-stage diagnosis, limited access to optimal treatment facilities, and socioeconomic disparities [2,3]. These factors may contribute to the observed trends, but the relationships are complex and influenced by a range of genetic, environmental, and healthcare system variables [4,5]. Projections suggest that breast cancer mortality rates in GCC countries will continue to rise if appropriate measures are not pursued to improve early detection, access to quality treatment, and patient outcomes; posing a significant public health challenge [6]. This knowledge can contribute to improved healthcare planning and resource allocation in the region.

In the GCC, breast cancer mortality, prevalence, and incidence are all on the rise, creating a rising public health concern. In addition to mortality, prevalence, and incidence, we included Disability-Adjusted Life Years (DALYs) to capture the overall burden of breast cancer by combining both fatal and non-fatal health outcomes. This metric provides a more comprehensive view of the disease's impact on population health and supports prioritization of healthcare interventions in the GCC region. Therefore, in this paper, our main goal was to assess the estimated burden of breast cancer using the Global Burden of Disease (GBD) outline. The GBD provided comprehensive and easily accessible epidemiological data on 369 diseases and injuries, as well as 87 risk variables. This comprehensive data comprises 7 super-regions, 21 regions, and more than 200 countries and territories [7]. By incorporating the population projections and age-standardized breast cancer incidence data from 1990

to 2019 and applying forecasting techniques, we aimed to provide perceptions into the forthcoming new cases of breast cancer in GCC countries [8].

This study is among the first to conduct a thorough, country-specific analysis of breast cancer trends using Global Burden of Disease (GBD) data across multiple Gulf Cooperation Council (GCC) countries. While GBD data are globally accessible, few studies have provided this level of focused insight into the temporal and spatial variation of breast cancer burden in the Gulf region over a nearly three-decade period. Understanding the current trends and forecasting future trajectories of breast cancer incidence, prevalence, DALYs, and mortality in these countries is crucial for implementing effective prevention planning and treatment strategies. These efforts should focus on promoting awareness, early detection through screening programs, improving access to quality healthcare services, and addressing sociocultural barriers to optimal care. By addressing these factors, GCC countries can make significant progress in reducing the burden of breast cancer and improving patient outcomes.

## Methodology

### Data source, acquisition, and study design

In the current systemic analysis of the GBD study, the primary data was obtained from GBD 2019. The GBD provided inclusive and accessible epidemiological data on 369 diseases and injuries, as well as 87 risk factors, from 1990 to 2019. This comprehensive data encompasses 7 super-regions, 21 regions, and more than 200 countries and territories, and was obtained using a rigorous methodology previously described [9,10]. To obtain the necessary data, we utilized the Global Health Data Exchange (GHDx) tool (https://ghdx.healthdata.org/gbd-2019). The study used the following measures: breast cancer incidence, prevalence, and mortality for each selected country. The metrics for all selected measures (incidence, prevalence, Disability-adjusted life-years "DALYs", and mortality) were the rate per 100,000 individuals. The study included the following risk factors that contributed to the burden of BC: dietary risks, low physical activity, metabolic risks, and tobacco. Although alcohol use is a known risk factor for breast cancer globally, in the GCC region, social, cultural, and legal factors result in significantly lower alcohol consumption compared to global averages [9–11]. Therefore, it was excluded from our analysis due to its low prevalence in this population. A detailed description of each risk factor's definition, selection and the estimation of uncertainty interval are explained elsewhere (https://www.who.int/standards/classifications/classification-of-diseases) [11]. The human development index (HDI) data at the national level were collected from the United Nations Development Programmer [12]. HDI is a measure that evaluates a country's progress in health, education, and standard of living. It offers a comprehensive view beyond economic growth, revealing a country's overall quality of life and opportunities, providing a comprehensive understanding of a nation's overall development.

### Study population

In this study, we included breast cancer population level data for Gulf Cooperation Council Countries (Saudi Arabia, Kuwait, the United Arab Emirates, Qatar, Bahrain, and Oman) from 1990 to 2019. The Global Burden of Disease study classified breast cancer according to the 10th revision of the International Classification of Diseases (ICD) 10 codes (i.e., C50-C50.9, D05-D05.9, D24-D24.9, D48.6, D49.3) [13]. This study includes both genders and utilized four intervals of age groups (20–24, 25–29, 30–34, 35–39, 40–44, 45–49, 50–54, 55–59, 60–64, 65–69, 70–74, 75–79, 80 + years old). The projections for the years 2020–2030 were obtained from the United Nations Department of Economic and Social Affairs/Population Division [14]. The population data was categorized by year, gender (both, female, male), and age (13 age groups spanning from 20 years old to 80 + years and older, in 5-year increments).

### Estimation of attributable burden

Estimating the attributable burden in GBD 2019 involves examining 560 pairs of risk-outcomes derived from systematically reviewed publications backed by credible or probable evidence. These 560 pairs are used for global estimates. The GBD

2019 utilized the Comparative Risk Assessment (CRA) method, extensively detailed in prior literature [9]. Within GBD 2019, four risk factors were identified as paired outcomes for breast cancer, including unhealthy diet (diet high in red meats), metabolic risk factors including both high body mass index (defined as BMI greater than 20–25 kg/m2), and elevated fasting plasma glucose (any level above 4.8–5.4 mmol/L), insufficient physical activity (< 3000–4500 metabolic equivalent (MET) minutes/week), and tobacco use (including smoking, and exposure to secondhand smoke). Data on the proportion of breast cancer deaths and DALYs attributable to these risk factors for six countries were extracted from GBD 2019.

### Data analysis

The incidence of BC by country was quantified using the estimated annual percentage change (EAPC) and its corresponding 95% uncertainty interval, as reported in the previous literature [15]. This approach allowed for a comprehensive assessment of the trends and variability in BC incidence over time and across different geographical locations. The Age-standardized rate (ASR) for breast cancer, already reported in the GBD data, were utilized in our analysis. We selected two-time points, specifically 1990 and 2019 to calculate the age-standardized incidence percentage change of BC for each selected risk factor, and country. The formula for calculating EAPC is as follows:

$$ASR = \frac{\sum_{i=0}^{A} aiwi}{\sum_{i=1}^{A} wi} \times 100,000$$

(ai, denotes the ith age class, and the number of persons (or weight) (wi) in the same age subgroup i of the selected reference standard population).

The trends in age-standardized rates serve as crucial indicators of evolving disease patterns and risk factors. To capture these trends over time, we employed the widely recognized ASR-EAPC measurement [16,17]. EAPC, developed to characterize ASR trends over defined time frames, operates under the assumption of a linear relationship between ASR and the natural logarithm.

In the equation $Y = \alpha + \beta X + \varepsilon$, Y represents the natural logarithm of the age-standardized rate "(ln (ASR)", X represents the calendar year, and $\in$ represents the error term. In this formula, the coefficient β determines whether ASR trends are positive or negative. The formula for calculating the EAPC is as follows:

$$EAPC = 100 \times [\exp(\beta) - 1].$$

The linear model offers 95% confidence intervals (CIs) [18,19]. When the EAPC and the lower CI are positive, the ASR tends to increase. Conversely, in a downward trend [16], ASR shows a negative EAPC and an upper CI. Projected numbers and ASRs of incidence until 2030 were extrapolated using the Bayesian age-period-cohort (BAPC) model [20,21]. Among several statistical methods evaluated, including the smooth spline model, and Poisson regression, the BAPC model emerged as the most suitable for projecting the cancer burden, particularly for short-term projections. Bayesian Poisson spatial model was employed to guarantee balanced estimations and precise results for every province. By using data from neighboring areas this spatial modeling technique makes it possible to produce accurate estimates for places with little or no data. All analyses were performed using SAS statistical software version 9.4 (SAS Institute Inc. Cary, NC).

### Results

#### Age standardized annual percentage change of breast cancer incidence, death, and DALY's in GCC countries from 1990 to 2019

Saudi Arabia had a significant increase in BC Age-Standardized Incidence Rates (ASIR) per 100,000 for both sexes (64.39%) and females (65.49%) from 1990 to 2019, accompanied by notable rises in Age-standardized death rate (ASDR)

(22.95%) and DALY's (24.12%) for both sexes, as well as for female ASDR (25.59%), and DALY's (24.6%). Oman had the highest increase in ASDR per 100,000 for females (31.17%), while Kuwait reported the highest decrease in ASDR for females (−36.28%) per 100,000 from 1990–2019. All countries experienced a decrease in male ASDR except for Bahrain where a 39.13% increase of male ASDR from 1990 to 2019 (Table 1).

### Sex and age distribution in prevalence, incidence, death, and DALY's rate in 2019

Significant disparities in the age and sex distribution of BC were observed in the counts of prevalence, incidence, death, and DALY's and the corresponding rate. In 2019, the IR, DR, and DALY's for males across all countries were higher in older age groups starting from 60 + compared to age groups starting from 35 + ages in females (Fig 1). The highest IR, and DR and DALY's for males were in Oman IR (42.52 per 100,000), and DR (30.22 per 100,000), and DALY's (344.4 per 100,000), and prevalence (239.3 per 100,000). In 2019, United Arab Emirates had the highest female ASIR (1839. per 100,000) DR (100.8 per 100,000), and DALY's (3409 per 100,000), and prevalence (218.4 per 100,000) in the age group 55–59 (Fig 1).

### Trends in ASIR, ASDR, prevalence and DALY's by sex from 1990 to 2019

Between 1990 and 2019, the ASIR, ASDR, prevalence, and DALYs of female and male BC increased greatly in all GCC countries. Among these countries, Saudi Arabia had the highest uptrend increase in ASIR among female (65.49%) and Bahrain had the highest ASIR in male (57.77%). In 2019, Qatar exhibited the highest ASIR (103.72 per 100,000; 95%CI 80.21,131.21). The highest uptrend increases in ASDR, and DALY's in female was in Oman (31.17%, 25.02% respectively), followed by Saudi Arabia (25.59%), while the highest ASDR uptrend increase was in Bahrain (39.13%). In 2019, the highest female ASDR was in Qatar (36.21 per 100,000; 95%UI 28.89,45.77) (Table 1, Fig 2). Oman exhibited the highest prevalence uptrend increases in male (9.41%), and Qatar exhibited the highest prevalence female uptrend increases (52.83%) over the study period (Table 1, S1 Table, Fig 2).

### Association of ASIR, ASDR, prevalence, and age standardized DALYs rate with HDI

There is a statistically significant positive correlation between ASIR and HDI across all countries with the highest correlation was in Saudi Arabia (r = 0.99; $p < .0001$). While a fluctuation was observed in the correlations between ASDR, DALY's and HDI across all countries. The highest significant negative correlation between ASDR and HDI was in Bahrain (r = −0.56; $p < 0.001$). The highest significant negative correlation between DALY's and HDI was in Qatar (r = −0.68; $p < .0001$) (Fig 3).

### BC ASDR attributable to risk factors

During 1990–2019, metabolic risk followed by tobacco consumption were the largest contributor of BC ASDR for females for all included countries. Metabolic risk contribution of BC ASDR was the highest in Qatar with (74.06%) in 1990, and (60.41%) in 2019. In 2019, tobacco contribution of BC ASDR for females was the highest in Kuwait (22.16%). Between 1990–2019, all GCC countries experienced an increase in metabolic risk contribution. The highest increase of metabolic risk was in Oman with (27.79%; 95% 27.1–28.5). Deaths associated with low physical activity; dietary risk displayed different trends across the included countries (Table 2, Fig 4).

### Predictions of BC incidence in CGG countries by sex

We compared incidence rate calculated by BAPC model with observed rate from the GBD 2019 between 1990 and 2019 and found that the highest BC projected new cases among females was in Saudi Arabia with (nearly 20%, 9441 new cases) increase in the ten years projection. Whereas Bahrain showed the highest projected BC new cases among males with approximately 21% increase (Table 3, Fig 5).

**Table 1. Age standardized annual percentage change of incidence, death, and DALY's burden rates in GCC countries from 1990 to 2019.**

| | ASIR[1,2] | | | ASDR[1,2] | | | DALYs[1,2] | | |
|---|---|---|---|---|---|---|---|---|---|
| | 1990 (95%UI)[3] | 2019 (95%UI) | Change % | 1990 (95%UI) | 2019 (95%UI) | Change, % | 1990 (95%UI) | 2019 (95%UI) | Change % |
| Saudi Arabia | | | | | | | | | |
| Both | 6.11 (4.47,8.25) | 17.16 (12.79,22.69) | 64.39 | 4.5 (3.27,6.22) | 5.84 (4.47, 7.6) | 22.95 | 134.91 (97.22,185.03) | 177.79 (132.71,177.79) | 24.12 |
| Female | 14.86 (10.75,20.12) | 43.06 (31.94,57.1) | 65.49 | 10.67 (7.72,14.83) | 14.34 (10.91,18.71) | 25.59 | 336.4 (241.47,461.45) | 446.14 (332.23,589.45 | 24.6 |
| Male | 0.32 (0.21,0.46) | 0.38 (0.25,0.54) | 15.78 | 0.27 (0.18,0.39) | 0.21 (0.13,0.29) | −28.57 | 6.28 (4.17,9.08) | 4.43 (2.98,6.24) | −41/76 |
| Kuwait | | | | | | | | | |
| Both | 15.52 (17.06,14.09) | 18.11 (14.6,23.15) | 14.3 | 7.06 (6.41,7.71) | 5.46 (4.45,6.91) | −29.30 | 194.58 (178.11,212.91) | 152.51 (124.01,195.61) | −27.59 |
| Female | 41.34 (37.56,45.76) | 42.76 (34.38,54.65) | 3.32 | 17.73 (16.13,19.39) | 13.01 (10.55,16.58) | −36.28 | 526.52 (481.84,578.84) | 358.84 (290.26,461.81) | −46.73 |
| Male | 0.5 (0.38,0.65) | 0.51 (0.35,0.75) | 1.96 | 0.32 (0.25,0.41) | 0.24 (0.17,0.35) | −33.33 | 6.78 (5.43,8.55) | 5.00 (3.59,7.24) | −36.6 |
| Qatar | | | | | | | | | |
| Both | 16.79 (12.55,22.51) | 25.45 (19.8,32.1) | 34.02 | 10.83 (7.94,14.71) | 8.98 (7.08,11.17) | −20.60 | 258.12 (195.57,341.36) | 209.71 (162.47,262.54) | −23.08 |
| Female | 48.92 (36.98,64.61) | 103.72 (80.21,131.21) | 52.83 | 28.21 (20.87,37.95) | 36.21 (28.89,45.77) | 22.09 | 797.43 (605.15,1045.64) | 856.37 (662.49.107.64) | 6.88 |
| Male | 0.76 (0.46,1.15) | 0.86 (0.52,1028) | 11.62 | 0.54 (0.33,0.81) | 0.42 (0.27,0.62) | −28.57 | 13.00 (7.96,19.27) | 8.35 (5.16,12.02) | −55.69 |
| Bahrain | | | | | | | | | |
| Both | 19.82 (16.86,23.22) | 26.15 (21.07,31.89) | 24.21 | 12.48 (10.72,14.44) | 10.61 (8.68,12.28) | −17.62 | 332.99 (283.43,390.33) | 253.44 (203.02,311.01) | −31.39 |
| Female | 46.76 (38.89,53.96) | 67.49 (54.04,83.05) | 30.71 | 27.38 (23.51,31.81) | 25.16 (20.44,30.45 | −8.82 | 788.57 (670.99,925.05) | 668.34 (533.29,827.71) | −17.99 |
| Male | 0.19 (0.13,0.26) | 0.45 (0.28,0.66) | 57.77 | 0.14 (0.1,0.2) | 0.23 (0.14,0.34) | 39.13 | 3.38 (2.44,4.58) | 5.05 (3.2,7.33) | 33.07 |
| Oman | | | | | | | | | |
| Both | 9.31 (6.62,13.32) | 19.47 (16.31,22.93) | 52.18 | 5.76 (4.11,8.21) | 7.65 (6.47,9.02) | 24.71 | 153.82 (109.11,220.59) | 187.98 (157.76.222.13) | 18.17 |
| Female | 19.28 (13.15,28.41) | 44.65 (36.83,52.87) | 56.82 | 10.93 (7.52,16.16) | 15.88 (13.29,18.91) | 31.17 | 326.03 (221.56,479.83) | 434.82 (359.14,519.01) | 25.02 |
| Male | 2.31 (1.46,3.37) | 2.54 (1.67,3.67) | 9.06 | 1.84 (1.17,2.66) | 1.42 (0.96,2.07) | −29.58 | 36.91 (23.65,52.78) | 26.27 (17.71,37.04) | −40.5 |
| United Arab Emirates | | | | | | | | | |
| Both | 13.98 (10.04,19.62) | 15.03 (11.5,19.41) | 6.99 | 9.38 (6.76,13.28) | 7.27 (5.61,9.31) | −29.02 | 245.35 (176.62,341.31) | 204.37 (156.21,266.21) | −20.05 |
| Female | 40.77 (28.7,57.79) | 57.47 (43.29,73.69) | 29.06 | 25.49 (17.94,36.31) | 26.19 (20.03,33.55) | 2.67 | 743.75 (526.28,1041.73) | 790.97 (593.96,1020.86) | 5.97 |
| Male | 1.21 (0.52,2.07) | 1.12 (0.52,2.04) | −8.04 | 0.96 (0.41,1.65) | 0.7 (0.33,1,24) | −37.14 | 21.86 (9.83,36.67) | 17.11 (8.07,30.67) | −27.76 |

[1] per 100,000.

[2] ASDR: Age-standardized death rate, ASIR: Age-standardized incidence rate, *DALYs:* Disability-adjusted life-years, BC: breast cancer.

[3] UI: Uncertainty interval.

| Age | 20-24 | 25-29 | 30-34 | 35-39 | 40-44 | 45-49 | 50-54 | 55-59 | 60-64 | 65-69 | 70-74 | 75-70 | 80+ |
|---|---|---|---|---|---|---|---|---|---|---|---|---|---|
| **Male** | | | | | | | | | | | | | |
| **United Arab Emirates** | | | | | | | | | | | | | |
| DR | 0.02 | 0.02 | 0.06 | 0.13 | 0.36 | 0.65 | 0.94 | 1.41 | 2.37 | 3.41 | 3.10 | 5.25 | 7.42 |
| DALYs | 1.40 | 1.55 | 3.57 | 7.14 | 17.50 | 28.24 | 36.27 | 47.44 | 69.00 | 84.14 | 62.84 | 84.33 | 79.4 |
| Prevalence | 0.70 | 0.89 | 1.61 | 2.87 | 7.10 | 11.73 | 14.72 | 20.24 | 32.64 | 43.91 | 39.07 | 50.50 | 55.2 |
| IR | 0.07 | 0.07 | 0.15 | 0.31 | 0.82 | 1.45 | 1.86 | 2.57 | 4.37 | 6.00 | 4.88 | 7.14 | 8.12 |
| **Saudi Arabia** | | | | | | | | | | | | | |
| DR | 0.01 | 0.01 | 0.01 | 0.02 | 0.05 | 0.11 | 0.17 | 0.27 | 0.49 | 0.82 | 1.07 | 2.15 | 3.40 |
| DALYs | 0.74 | 0.70 | 0.70 | 1.14 | 2.65 | 4.84 | 6.51 | 9.38 | 14.54 | 20.63 | 22.24 | 35.24 | 35.5 |
| Prevalence | 0.46 | 0.49 | 0.48 | 0.70 | 1.64 | 2.91 | 3.76 | 5.71 | 9.93 | 15.91 | 18.54 | 27.14 | 29.1 |
| IR | 0.05 | 0.04 | 0.04 | 0.06 | 0.17 | 0.32 | 0.42 | 0.63 | 1.14 | 1.85 | 2.11 | 3.63 | 4.23 |
| **Kuwait** | | | | | | | | | | | | | |
| DR | 0.01 | 0.01 | 0.01 | 0.01 | 0.07 | 0.10 | 0.17 | 0.21 | 0.66 | 1.54 | 1.17 | 1.56 | 4.68 |
| DALY's | 0.72 | 0.67 | 0.61 | 0.56 | 3.44 | 4.48 | 6.90 | 7.23 | 19.63 | 38.95 | 24.63 | 25.93 | 40.4 |
| Prevalence | 0.54 | 0.55 | 0.51 | 0.53 | 2.53 | 3.38 | 4.81 | 5.54 | 15.63 | 33.69 | 24.29 | 26.46 | 30.7 |
| IR | 0.06 | 0.05 | 0.04 | 0.04 | 0.27 | 0.37 | 0.53 | 0.57 | 1.83 | 4.11 | 2.74 | 3.09 | 5.09 |
| **Qatar** | | | | | | | | | | | | | |
| DR | 0.01 | 0.01 | 0.03 | 0.07 | 0.16 | 0.28 | 0.48 | 0.93 | 1.60 | 1.60 | 1.77 | 4.51 | 7.64 |
| DALY's | 0.69 | 0.67 | 1.59 | 3.31 | 7.33 | 11.10 | 16.59 | 27.78 | 40.42 | 40.42 | 36.92 | 74.01 | 89.32 |
| Prevalence | 0.65 | 0.67 | 1.24 | 2.64 | 5.37 | 7.67 | 12.18 | 22.48 | 36.75 | 36.75 | 37.55 | 66.82 | 84.18 |
| IR | 0.05 | 0.04 | 0.10 | 0.25 | 0.57 | 0.83 | 1.31 | 2.54 | 4.22 | 4.22 | 4.04 | 8.61 | 11.66 |
| **Oman** | | | | | | | | | | | | | |
| DR | 0.01 | 0.01 | 0.03 | 0.08 | 0.23 | 0.49 | 0.80 | 1.39 | 2.38 | 5.15 | 6.26 | 14.95 | 30.22 |
| DALY's | 0.85 | 0.97 | 1.72 | 4.44 | 11.50 | 21.79 | 31.25 | 47.31 | 69.92 | 128.8 | 129.2 | 241.5 | 344.4 |
| Prevalence | 0.73 | 1.01 | 1.49 | 2.93 | 7.36 | 13.51 | 18.42 | 28.87 | 47.43 | 92.22 | 100.6 | 169 | 239.3 |
| IR | 0.06 | 0.06 | 0.10 | 0.26 | 0.75 | 1.52 | 2.11 | 3.33 | 5.71 | 11.85 | 13.00 | 26.00 | 42.52 |
| **Bahrain** | | | | | | | | | | | | | |
| DR | 0.01 | 0.01 | 0.01 | 0.01 | 0.08 | 0.12 | 0.12 | 0.18 | 0.36 | 1.25 | 1.80 | 4.00 | 2.20 |
| DALY's | 0.81 | 0.76 | 0.67 | 0.82 | 3.76 | 5.15 | 4.71 | 5.99 | 10.64 | 31.25 | 36.96 | 64.95 | 25.0 |
| Prevalence | 0.50 | 0.52 | 0.48 | 0.60 | 2.24 | 3.17 | 3.05 | 4.18 | 7.98 | 23.00 | 28.23 | 44.52 | 24.7 |
| IR | 0.05 | 0.05 | 0.04 | 0.05 | 0.24 | 0.35 | 0.31 | 0.40 | 0.84 | 2.82 | 3.54 | 6.67 | 2.94 |
| **Female** | | | | | | | | | | | | | |
| **United Arab Emirates** | | | | | | | | | | | | | |
| DR | 0.10 | 0.61 | 4.07 | 10.85 | 22.65 | 42.18 | 74.13 | 100.8 | 87.81 | 88.14 | 104.7 | 110.8 | 146.8 |
| DALY's | 7.06 | 40.1 | 242.5 | 587.6 | 1111 | 1847. | 2861. | 3409. | 2559. | 2175. | 2117. | 1782. | 1712. |
| Prevalence | 5.43 | 28.7 | 146.8 | 336.8 | 705. | 1101. | 1517. | 1839. | 1689. | 1699. | 1703. | 1763. | 2090. |
| IR | 0.57 | 3.19 | 17.06 | 39.65 | 82.97 | 129.8 | 181.8 | 218.4 | 184.8 | 173.9 | 169.9 | 165.4 | 199.5 |
| **Saudi Arabia** | | | | | | | | | | | | | |
| DR | 0.14 | 0.70 | 3.29 | 8.11 | 15.22 | 25.33 | 35.46 | 42.85 | 45.95 | 51.22 | 60.10 | 72.24 | 87.65 |
| DALYs | 9.90 | 46.8 | 200.5 | 448.0 | 764.4 | 1131. | 1392. | 1470. | 1363. | 1286. | 1229. | 1182. | 926.5 |
| Prevalence | 10.1 | 46.6 | 174.1 | 361.8 | 692.7 | 928.4 | 1010. | 1100. | 1204. | 1316. | 1238. | 1370. | 1309. |
| IR | 1.14 | 5.36 | 20.20 | 42.29 | 80.87 | 107.9 | 116.1 | 121.5 | 125.9 | 132.9 | 121.5 | 135.0 | 131.3 |
| **Kuwait** | | | | | | | | | | | | | |
| DR | 0.16 | 0.54 | 1.76 | 4.02 | 8.40 | 16.50 | 22.28 | 36.27 | 42.13 | 55.58 | 62.88 | 70.45 | 137. |
| DALY's | 11.6 | 37.2 | 109. | 227. | 430. | 750.9 | 887.9 | 1260. | 1268. | 1418. | 1309. | 1169. | 1333. |
| Prevalence | 14.9 | 46.9 | 122. | 238. | 508. | 807.3 | 859.9 | 1241. | 1507. | 1885. | 1761. | 1794. | 1914 |
| IR | 1.68 | 5.32 | 13.8 | 26.9 | 57.3 | 90.22 | 91.76 | 126.5 | 142.0 | 174.8 | 153.5 | 155.0 | 211. |
| **Qatar** | | | | | | | | | | | | | |
| DR | 0.24 | 0.51 | 3.91 | 9.33 | 16.20 | 27.09 | 40.41 | 67.74 | 99.63 | 135.3 | 165.1 | 291.3 | 545.8 |
| DALYs | 17.8 | 34.7 | 242.2 | 524.1 | 827.4 | 1228. | 1603. | 2336. | 2973. | 3418. | 3398. | 4766. | 6780. |
| Prevalence | 22.3 | 43.3 | 258.3 | 528.2 | 933.0 | 1284. | 1476. | 2116. | 2987. | 3750. | 3509. | 5137. | 7861. |
| IR | 2.51 | 4.81 | 29.50 | 60.43 | 106.0 | 143.6 | 160.8 | 229.1 | 326.0 | 416.7 | 393.7 | 634.4 | 1108. |
| **Oman** | | | | | | | | | | | | | |
| DR | 0.12 | 0.55 | 2.29 | 5.14 | 9.25 | 15.60 | 27.53 | 44.10 | 54.98 | 73.82 | 92.51 | 113.3 | 116. |
| DALY's | 8.84 | 37.32 | 140.6 | 286.2 | 466.7 | 700.0 | 1082. | 1515. | 1633. | 1852. | 1895. | 1862. | 1306. |
| Prevalence | 9.58 | 39.46 | 128.4 | 249.5 | 454.5 | 641.7 | 880.6 | 1220. | 1512. | 1872. | 1837. | 2047. | 1854. |
| IR | 1.07 | 4.49 | 14.74 | 28.64 | 51.70 | 71.00 | 95.68 | 131.4 | 159.7 | 199.9 | 196.6 | 220.8 | 188. |
| **Bahrain** | | | | | | | | | | | | | |
| DR | 0.22 | 0.75 | 4.33 | 10.52 | 18.37 | 10.52 | 4.33 | 0.75 | 0.22 | 86.53 | 115.3 | 193.7 | 265.7 |
| DALY's | 15.86 | 50.67 | 264.5 | 581.9 | 922.7 | 581.9 | 264.5 | 50.67 | 15.86 | 2163. | 2349. | 3146. | 3060. |
| Prevalence | 16.25 | 50.91 | 230.9 | 478.5 | 853.4 | 478.5 | 230.9 | 50.91 | 16.25 | 2135. | 2145. | 2900. | 3179. |
| IR | 1.83 | 5.78 | 26.64 | 55.47 | 98.28 | 55.47 | 26.64 | 5.78 | 1.83 | 225.2 | 236.3 | 362.5 | 439. |

DR: death rate, IR: incidence rate, DALYs: Disability-adjusted life-years

**Fig 1. Decomposition of age for prevalence, IR, DR, and DALY's rate in 2019 by sex (Heatmap).**

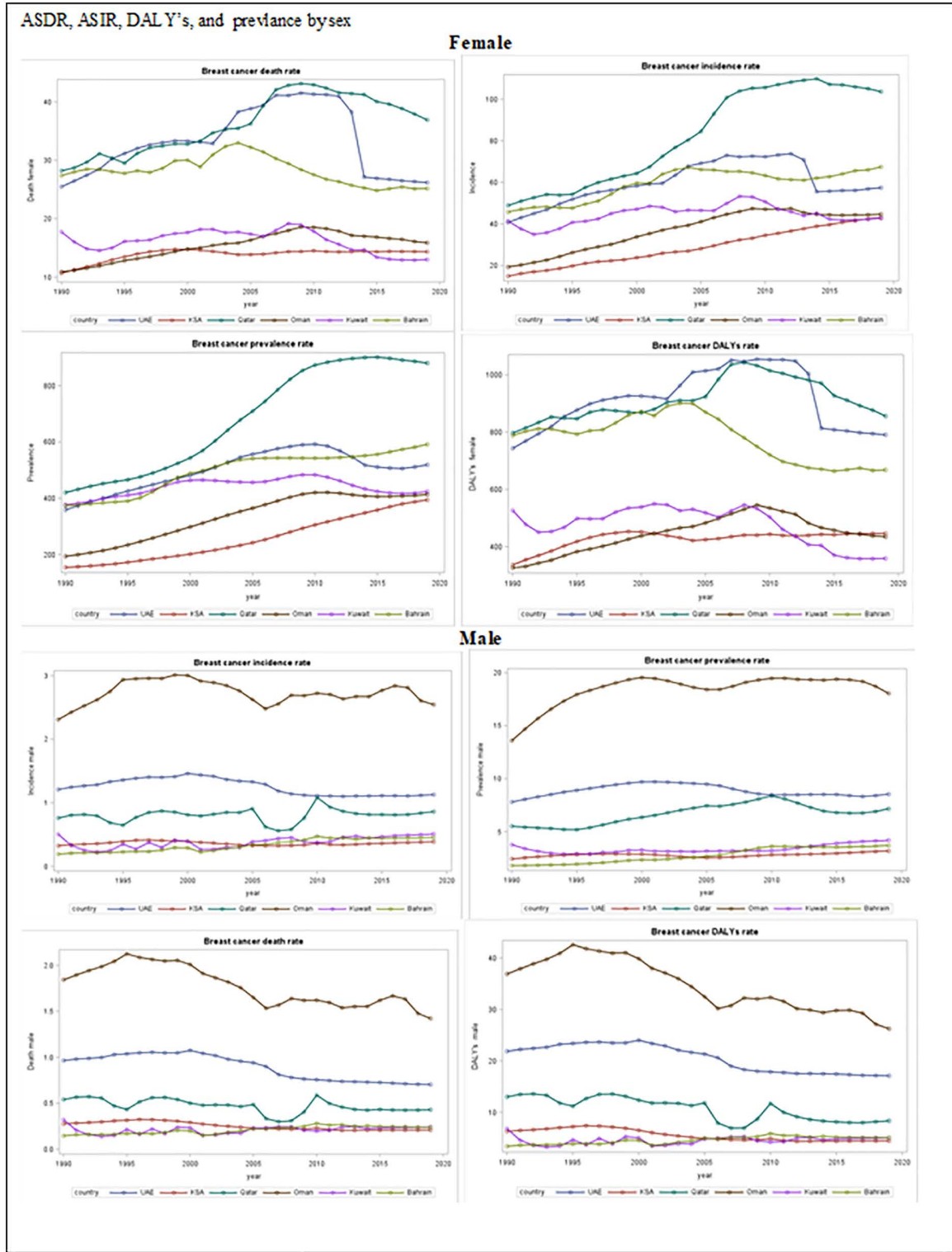

**Fig 2. Temporal trends for age standardized DALY's, ASIR, ASDR and prevalence by sex (1990 to 2019).**

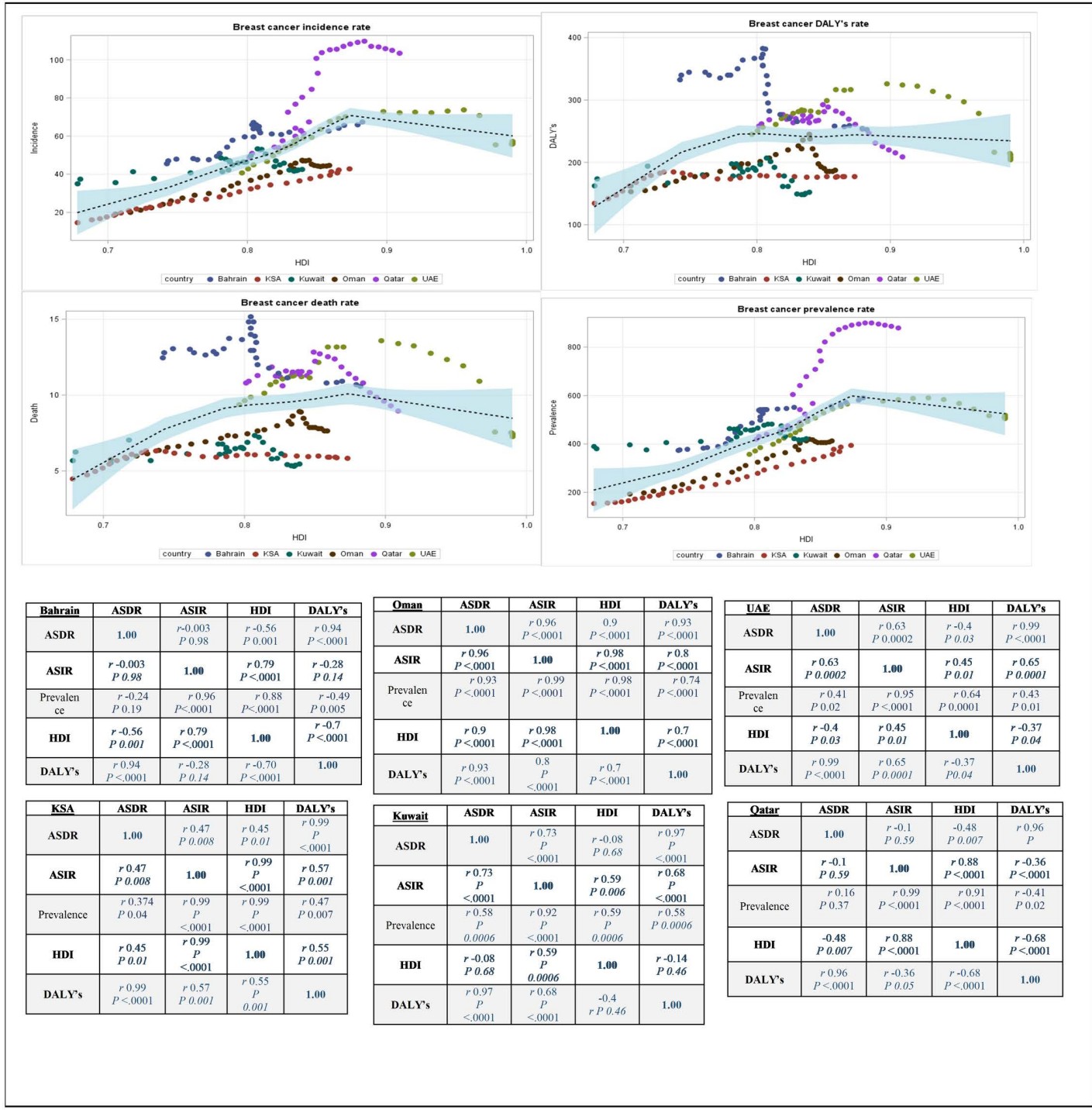

**Fig 3. Spearman correlation between ASIR, ASDR, DALY's, and prevalence with HDI.**

**Table 2. Age standardized percentage changes in breast cancer death attributable to risk factors from 1990 and 2019 (female)[5].**

| Risk factors | Saudi Arabia (95% UI)[1] | United Arab Emirates (95% UI) | Qatar (95% UI) | Kuwait (95% UI) | Oman (95% UI) | Bahrain (95% UI) |
|---|---|---|---|---|---|---|
| **1990** | | | | | | |
| Dietary risks[2] | 0.13 (0.00002,0.27) | 0.13 (−0.00007, 0.28) | 0.13 (−0.00004, 0.28) | 0.13 (−0.00006 −0.28) | 0.13 (−0.00004,0.28) | 0.13 (−0.00005 −0.28) |
| Low physical Activity | 0.03 (0.0005, 0.005) | 0.028 (0.005, 0.05) | 0.036 (0.007, 0.064) | 0.044 (0.008, 0.07) | 0.031 (0.006, 0.056) | 0.03 (0.006, 0.058) |
| Metabolic risks[3] | 0.08 (−0.01,0.17) | 0.11 (−0.01,0.25) | 0.13 (−0.02,0.26) | 0.13 (−0.017, 0.27) | 0.088 (−0.011,0.18) | 0.12 (−0.017, 0.25) |
| Tobacco | 0.014 (0.0005, 0.027) | 0.02 (0.001, 0.04) | 0.016 (−0.0001,0.03) | 0.027 (0.0023, 0.053) | 0.017 (0.0025,0.032) | 0.024 (0.006, 0.042) |
| **2019** | | | | | | |
| Dietary risks | 0.12 (−0.00006,0.27) | 0.13 (−0.00003, 0.28) | 0.13 (−0.00004, 0.28 | 0.13 (−0.00006 −0.28) | 0.13 (−0.00004,0.28) | 0.13 (−0.00005 - 0.28) |
| Low physical activity | 0.03 (0.0006, 0.005) | 0.032 (0.006, 0.061) | 0.036 (0.007, 0.064) | 0.044 (0.008, 0.07) | 0.031 (0.006, 0.056) | 0.033 (0.006, 0.059) |
| Metabolic risks | 0.14 (−0.02, 0.28) | 0.2 (−0.0310.39) | 0.2 (−0.03,0.38) | 0.18 (−0.025, 0.34) | 0.15 (−0.02, 0.31) | 0.17 (−0.029, 0.34) |
| Tobacco[4] | 0.016 (0.0002, 0.03) | 0.015 (0.0007,0.03) | 0.016 (−0.0001,0.03 | 0.022 (0.0007, 0.044) | 0.014 (0.0017, 0.027) | 0.021 (0.0035, 0.037) |
| **Changes in percentage from 1990 to 2019** | | | | | | |
| Dietary risks | −7.69 (−7.710 − −7.67) | ----[6] | ---- | ---- | ---- | ---- |
| Low physical activity | ---- | 14.28 (14.26–14.3) | ---- | ---- | ---- | ---- |
| Metabolic risks | 75 (74.98–75.02) | 81.81 (81.79–81.83) | 53.84 (53.82–53.86) | 14.28 (14.26–14.3) | 87.5 (87.48–87.52) | 41.66 (41.64–41.68) |
| Tobacco | 14.28 (14.26–14.3) | −25 (−25.02−−24.98) | ---- | −22.72 (−22.74 − −22.7) | −21.42 (−21.44 − −21.4) | −14.28 (−14.3 − −14.26) |

[1]Uncertainty interval (UI).

[2]Dietary risks refer to diet high in red meat.

[3]Metabolic risk factors include high fasting plasma glucose (FPG) and high body mass index (BMI).

[4]Tobacco include smoking and secondhand smoke.

[5]Dietary risks, and Metabolic risk are not available for male.

[6]no percentage change.

## Discussion

The current investigation highlights gender-specific trends in mortality and incidence, such as substantial increase in ASIR for breast cancer among females in Saudi Arabia from 1990 to 2019. Regarding ASDR per 100,000 in females, Oman experienced the highest increase, while Kuwait experienced the highest decrease. Bahrain saw a significant increase in male ASDR between 1990 and 2019, in contrast to all other countries where male ASDR decreased over this period. The study also establishes a positive correlation between the Human Development Index (HDI) and ASIR, highlighting the impact of socioeconomic development on breast cancer burden. Metabolic risk and tobacco use were identified as the primary contributors to ASDR of breast cancer in all studied countries, enhancing understanding of modifiable risk factors and adding policy relevance by pointing to preventable drivers of disease burden in the region. The ten-year BC prediction indicates a significant increase in female cases, with Saudi Arabia expected to experience the highest rise. Understanding

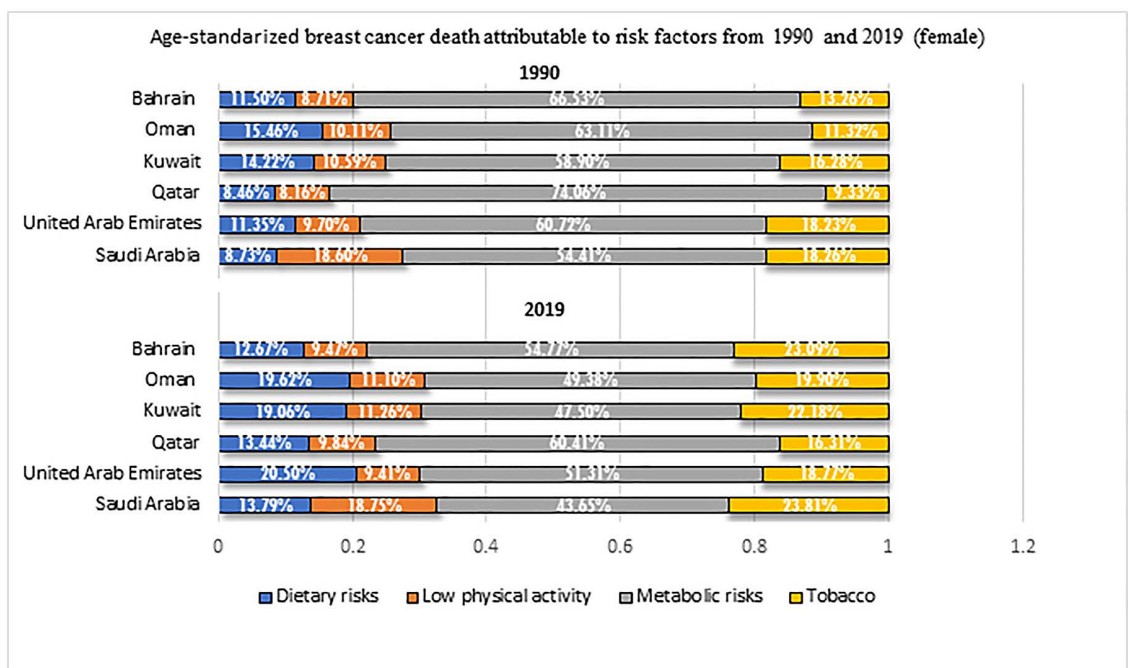

**Fig 4. Age-standardized percentage changes in breast cancer death attributable to risk factors from 1990 and 2019 (female).**

current trends and anticipating future breast cancer incidence, prevalence, and mortality in these countries is critical for developing effective prevention and treatment plans. The novelty lies in the integration of longitudinal epidemiological data, regional specificity, risk factor attribution, and predictive modeling, delivering a powerful tool for policymakers, researchers, and clinicians in the Gulf.

## ASIR trends in women

The results of this study show a significant increase in ASIR for breast cancer among females and both sex in Saudi Arabia between 1990 and 2019. This finding reflects the collaborative efforts of the Saudi government and healthcare organizations to offer easily accessible screening services and advocate for regular mammograms to facilitate early detection [22,23]. Also, Saudi Arabia established a National Plan Cancer Control to ensure early detection and reduction of cancer-related mortality rates. This initiative employs a variety of strategies, including improving access to diagnostic and treatment services, improving healthcare infrastructure, and conducting public awareness campaigns. The National Plan for Cancer Control demonstrates the government's commitment to reducing the cancer burden and improving patient outcomes across the country [24].

   Another finding is that Oman experienced the highest increase in ASDR per 100,000 for females (31.17%), while Kuwait experienced the highest decrease (−36.28%) per 100,000 from 1990–2019. These significant variations in ASDR among females reflect the challenges and success in cancer control efforts in the GCC region. Further research is required to assess the various factors influencing healthcare systems and public health initiatives in Oman, Kuwait, and other GCC countries, such as healthcare infrastructure, access to healthcare services, advances in diagnostic capabilities, and changes in lifestyle factors or environmental exposures that contribute to higher mortality rates. This variation in ASDR between Kuwait and Oman could be attributed to the fact that Kuwait has a better health system capacity than Oman as explained by The World Health Organization (WHO). WHO's report on Kuwait's cancer patient

**Table 3. Age standardized projection of new cases of breast cancer in GCC countries from 2020 to 2030[3.]**

| Year | United Arab Emirates | | | Saudi Arabia | | | Kuwait | | | Oman | | | Bahrain | | | Qatar | | |
|---|---|---|---|---|---|---|---|---|---|---|---|---|---|---|---|---|---|---|
| | Value1 | 95%CI2 | | value | 95%CI | | value | 95%CI | | value | 95%CI | | value | 95%CI | | value | 95%CI | |
| Female | | | | | | | | | | | | | | | | | | |
| 2020 | 56.82 | 50.29 | 63.35 | 44.06 | 43.46 | 44.66 | 43.20 | 39.36 | 47.05 | 44.83 | 43.58 | 46.08 | 68.64 | 66.06 | 71.22 | 102.26 | 98.78 | 105.73 |
| 2021 | 56.17 | 46.17 | 66.16 | 45.05 | 44.20 | 45.89 | 43.64 | 35.83 | 51.45 | 45.00 | 42.58 | 47.41 | 69.79 | 65.00 | 74.57 | 100.79 | 93.03 | 108.56 |
| 2022 | 55.51 | 42.31 | 68.71 | 46.04 | 45.00 | 47.07 | 44.08 | 31.59 | 56.58 | 45.17 | 41.40 | 48.93 | 70.93 | 63.65 | 78.22 | 99.33 | 86.34 | 112.32 |
| 2023 | 54.85 | 38.50 | 71.21 | 47.03 | 45.83 | 48.23 | 44.52 | 26.7 | 62.34 | 45.34 | 40.05 | 50.62 | 72.08 | 62.01 | 82.15 | 97.87 | 78.86 | 116.89 |
| 2024 | 54.20 | 34.65 | 73.75 | 48.02 | 46.68 | 49.36 | 44.96 | 21.23 | 68.69 | 45.51 | 38.55 | 52.46 | 73.22 | 60.10 | 86.35 | 96.41 | 70.67 | 122.15 |
| 2025 | 53.54 | 30.73 | 76.35 | 49.01 | 47.54 | 50.48 | 45.4 | 15.23 | 75.57 | 45.67 | 36.90 | 54.45 | 74.37 | 57.95 | 90.79 | 94.95 | 61.84 | 128.06 |
| 2026 | 52.89 | 26.73 | 79.04 | 50.00 | 48.42 | 51.59 | 45.84 | 8.75 | 82.93 | 45.84 | 35.12 | 56.57 | 75.52 | 55.57 | 95.47 | 93.49 | 52.43 | 134.56 |
| 2027 | 52.23 | 22.63 | 81.83 | 50.99 | 49.30 | 52.69 | 46.28 | 1.81 | 90.75 | 46.01 | 33.21 | 58.81 | 76.66 | 52.96 | 100.36 | 92.03 | 42.46 | 141.6 |
| 2028 | 51.57 | 18.44 | 84.71 | 51.98 | 50.18 | 53.78 | 46.72 | −5.56 | 99.00 | 46.18 | 31.19 | 61.18 | 77.81 | 50.15 | 105.46 | 90.57 | 31.98 | 149.16 |
| 2029 | 50.92 | 14.15 | 87.69 | 52.97 | 51.08 | 54.87 | 47.15 | −13.35 | 107.66 | 46.35 | 29.05 | 63.65 | 78.96 | 47.15 | 110.76 | 89.11 | 21. | 157.2 |
| 2030 | 50.26 | 9.76 | 90.77 | 53.96 | 51.97 | 55.96 | 47.59 | −21.52 | 116.71 | 46.52 | 26.81 | 66.23 | 80.1 | 43.96 | 116.25 | 87.65 | 9.59 | 165.71 |
| Male | | | | | | | | | | | | | | | | | | |
| 2020 | 1.14 | 1.08 | 1.19 | 0.39 | 0.37 | 0.41 | 0.49 | 0.36 | 0.62 | 2.47 | 2.30 | 2.63 | 0.46 | 0.42 | 0.51 | 0.85 | 0.64 | 1.05 |
| 2021 | 1.14 | 1.05 | 1.24 | 0.40 | 0.36 | 0.44 | 0.50 | 0.37 | 0.63 | 2.39 | 2.08 | 2.69 | 0.47 | 0.41 | 0.54 | 0.85 | 0.64 | 1.06 |
| 2022 | 1.15 | 1.01 | 1.29 | 0.41 | 0.35 | 0.46 | 0.51 | 0.38 | 0.64 | 2.31 | 1.85 | 2.77 | 0.48 | 0.40 | 0.56 | 0.85 | 0.65 | 1.06 |
| 2023 | 1.16 | 0.96 | 1.35 | 0.41 | 0.33 | 0.49 | 0.51 | 0.38 | 0.64 | 2.23 | 1.60 | 2.86 | 0.49 | 0.40 | 0.58 | 0.85 | 0.65 | 1.06 |
| 2024 | 1.16 | 0.91 | 1.41 | 0.42 | 0.32 | 0.52 | 0.52 | 0.39 | 0.65 | 2.15 | 1.34 | 2.96 | 0.50 | 0.40 | 0.61 | 0.86 | 0.65 | 1.06 |
| 2025 | 1.17 | 0.86 | 1.48 | 0.42 | 0.30 | 0.55 | 0.53 | 0.40 | 0.66 | 2.07 | 1.06 | 3.08 | 0.51 | 0.40 | 0.63 | 0.86 | 0.65 | 1.07 |
| 2026 | 1.18 | 0.80 | 1.56 | 0.43 | 0.28 | 0.58 | 0.53 | 0.40 | 0.66 | 1.99 | 0.77 | 3.21 | 0.52 | 0.40 | 0.65 | 0.86 | 0.66 | 1.07 |
| 2027 | 1.19 | 0.74 | 1.64 | 0.44 | 0.26 | 0.61 | 0.54 | 0.41 | 0.67 | 1.91 | 0.47 | 3.36 | 0.54 | 0.41 | 0.66 | 0.86 | 0.66 | 1.07 |
| 2028 | 1.19 | 0.67 | 1.72 | 0.44 | 0.24 | 0.65 | 0.55 | 0.42 | 0.68 | 1.83 | 0.16 | 3.51 | 0.55 | 0.41 | 0.68 | 0.87 | 0.66 | 1.07 |
| 2029 | 1.20 | 0.60 | 1.80 | 0.45 | 0.21 | 0.68 | 0.55 | 0.43 | 0.68 | 1.76 | −0.17 | 3.68 | 0.56 | 0.41 | 0.7 | 0.87 | 0.66 | 1.08 |
| 2030 | 1.21 | 0.53 | 1.89 | 0.45 | 0.19 | 0.72 | 0.56 | 0.43 | 0.69 | 1.68 | −0.51 | 3.86 | 0.57 | 0.41 | 0.72 | 0.87 | 0.67 | 1.08 |

[1]precited value.

[2]95% confidence interval.

[3]values per 100,000.

care reveals high-quality patient care, with a population-based cancer registry (PBCR) of 11.2 available per 10,000 cancer patients. The report also shows a high availability of external beam radiotherapy units (photon and electron) and diagnostic imaging capabilities, with 273.6 CT and 234.5 MRI scanners per 10,000 cancer patients, and 30.7 PET or PET/CT scanners per 10,000 cancer patients [25]. In contrast, there was significantly less medical equipment in Oman that was needed for cancer detection and treatment. Oman had 45.2 mammographs, 138.5 CT scanners, 111.4 MRI scanners, and 6 PET or PET/CT scanners per 10,000 cancer patients [26]. While specific comparative data on medical resource distribution are limited, the availability and accessibility of comprehensive cancer treatment, including specialized oncology services and advanced medical technologies, are crucial for improving patient outcomes. Differences in these resources between Oman and Kuwait may influence ASDR trends, possibly due to variations in healthcare infrastructure and resource allocation. Therefore, this disparity suggests that Kuwaiti patients may have better access to essential radiotherapy services, potentially leading to improved cancer outcomes. Furthermore, Kuwait has established organized screening programs aimed at early cancer detection. The Kuwait National Mammography Screening Program (KNMSP), initiated in 2014, offers nationwide breast cancer screening for women aged 40 and above. Over five years, the program screened 14,773 women, which facilitated early treatment and likely contributed to the observed decrease in ASDR among Kuwaiti women [27].

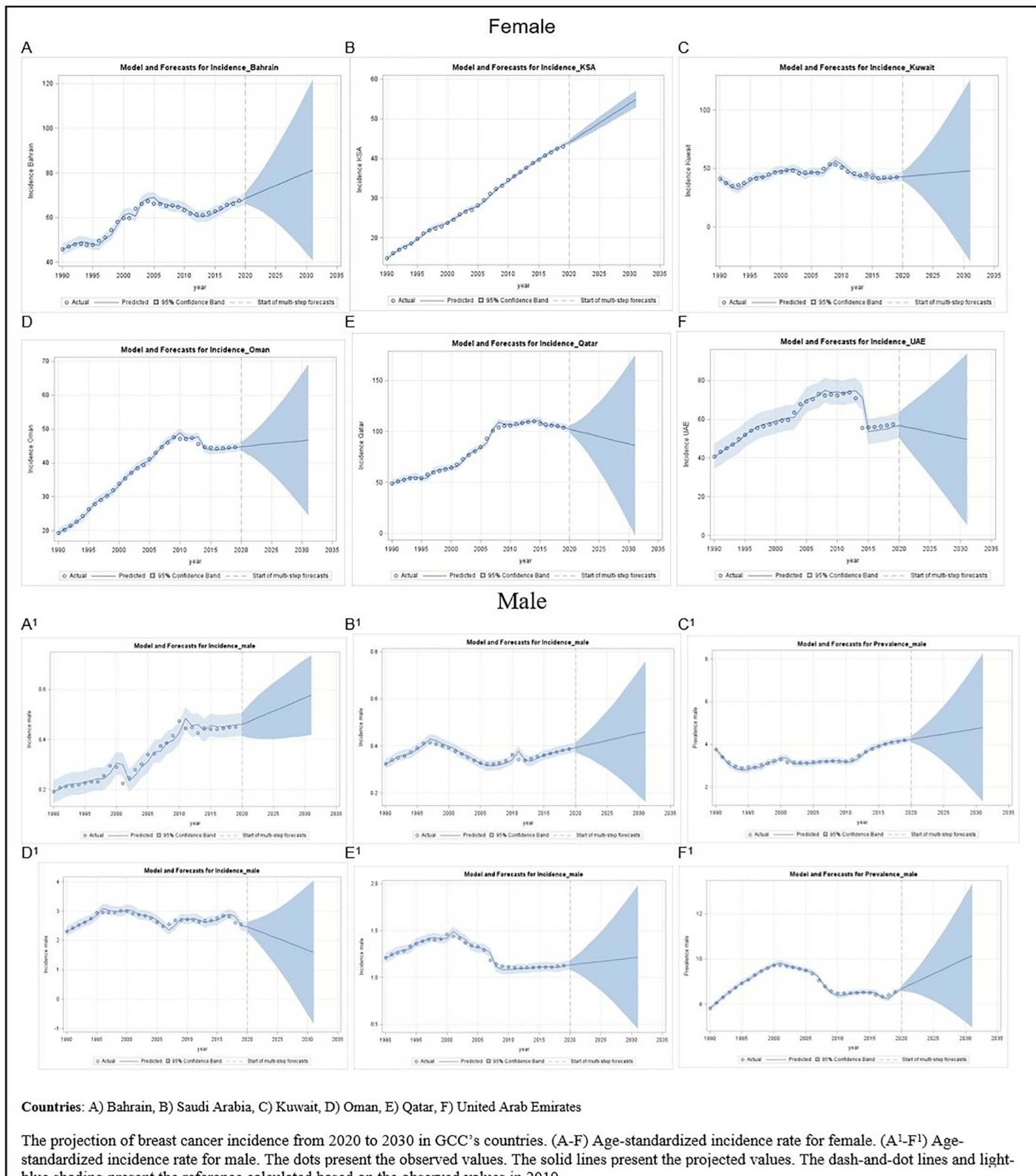

**Countries:** A) Bahrain, B) Saudi Arabia, C) Kuwait, D) Oman, E) Qatar, F) United Arab Emirates

The projection of breast cancer incidence from 2020 to 2030 in GCC's countries. (A-F) Age-standardized incidence rate for female. (A[1]-F[1]) Age-standardized incidence rate for male. The dots present the observed values. The solid lines present the projected values. The dash-and-dot lines and light-blue shading present the reference calculated based on the observed values in 2019.

**Fig 5. ASIR prediction from 2020 to 2030 by sex.**

## ASDR trends in men

Bahrain experienced a high increase in male ASDR (39.13%), whereas all other GCC countries reported declines in male ASDR between 1990 and 2019. The available research on male breast cancer is quite limited. A single study of occult triple-negative male breast cancer was reported from Bahrain [28]. This finding was unexpected and necessitated more inquiry to determine why Bahrain has a different pattern in male ASDR compared to the other GCC states. The finding could be attributed to differences in awareness, healthcare infrastructure, screening differences, public attitudes, and data reporting processes between nations. Another explanation for this finding is that Bahrain's healthcare expenditures are higher than the GCC regional average. In addition, unlike neighboring countries, Bahraini physicians and nurses constitute the majority of the country's healthcare personnel [27]. Moreover, the UN World Population Prospects data show that Bahrain's population structure is shifting with increasing proportions of older adults. Therefore, although ASDR adjusts for age, Bahrain's relatively aging male population and longer life expectancy may contribute to a higher ASCDR [29]. In addition, Bahrain exhibits high rates of obesity, diabetes, and hypertension, which are significant risk factors for various cancers. The rising prevalence of these conditions may also contribute to increased cancer mortality [29].

Our Data demonstrated that male breast cancer occurrences among people over the age of 60 in studied counties have increased significantly in recent decades, however, research in this field remains restricted. A substantial correlation between negative emotions and heightened risk of breast cancer initiation was uncovered in a recent study, with indications that these emotions could also impact the prognosis of the disease [30]. The rising frequency of depression among the elderly may be connected to an increase in male BC occurrences [31]. Despite its low incidence rate, our findings highlight the urgent need for increased emphasis on male BC in the GCC area. Age- and sex-specific DALY patterns reveal critical disparities: male DALYs are concentrated in older age groups (60+), with Oman and the UAE showing the highest respective rates, while in females the elevated IR, DR, and DALYs emerges from the age group of 35 years and above. This could be due to improved awareness in the younger generation with the availability of social media, marketing for BC campaigns, and increased literacy compared to the older female generation. In UAE, most of the BC cases occurred in women under the age of 50. In a recent study, death rate dropped after the age of 65, this could be explained by the low and young population of the UAE as well as illiteracy in older patients [29].

## Association between ASIR and HDI

The statistically substantial positive connection between ASIR and HDI, which is observed in all countries, points to an intriguing relationship between socioeconomic development and the incidence of breast cancer. This result suggests that as countries progress in terms of human development, which can be measured by factors like income, education, and access to healthcare, the incidence rates of not only breast cancer but also other diseases like kidney cancer [32] and chronic respiratory disorders [33] tend to grow. This association likely reflects complex interactions among lifestyle changes, reproductive behaviors, and healthcare access that accompany socioeconomic progress. For example, higher HDI is often associated with delayed childbearing, fewer pregnancies, and increased use of hormonal therapies—all factors correlated with breast cancer risk [4,5]. Moreover, diets in higher-HDI settings tend to include more processed and high-fat foods, which have been associated with increased breast cancer risk in observational studies [34]. Urbanization and sedentary lifestyles, which are linked to better socioeconomic position and common in more developed regions, contribute to metabolic changes such as obesity and insulin resistance, which are linked to breast cancer through hormonal and inflammatory [35–37]. Furthermore, higher-HDI countries may have better screening systems and healthcare infrastructure, which might result in increased reported incidence by enabling earlier and more frequent detection [38]. This phenomenon is reflected globally, with high-income countries reporting breast cancer incidence rates exceeding 80 per 100,000, while low-income countries report rates below 40 per 100,000 [39]. BC is a major public health problem, especially in low-income countries that have limited capacity to provide breast cancer programs comparable to high-income countries [40,41]. While ASIR rises with HDI, we also observed fluctuations in correlations between ASDR, DALYs, and

HDI across countries highlight the complex interplay of factors such as healthcare infrastructure, public health initiatives, socioeconomic inequalities, and cultural influences [42–44]. These findings emphasize the necessity of incorporating socioeconomic indicators like HDI when designing breast cancer prevention, screening, and treatment strategies. Tailored interventions that address country-specific health system strengths and gaps are essential to reduce disparities in breast cancer outcomes both within and between GCC countries.

## Metabolic and lifestyle factors in breast cancer: evidence and insights

Our data show that metabolic risk including elevated body mass index and high fasting plasma glucose—was the main attributable of ASDR of breast cancer in the GCC. These metabolic conditions may promote carcinogenesis via hormonal dysregulation, chronic inflammation, and altered adipokine profiles [45–47]. This finding is consistent with Smolarz's research (2022), which highlights the increase in breast cancer incidence in developed countries as a result of lifestyle variables such poor eating habits, sedentary lifestyles, and stress [41]. Additionally, our results suggest that the second attributed factor of ASDR of breast cancer among females in all studied countries from 1990 to 2019 was tobacco use including both active smoking and secondhand smoke exposure. These associations align with findings from recent research identifying smoking as a major cause of breast cancer in North African and Middle Eastern nations [48]. Tobacco use, correlates with increased breast cancer risk and worse outcomes, potentially through DNA damage and immune modulation [49,50]. Other well-established risk factors—such as reproductive history (early menarche, late menopause, parity), breastfeeding practices, and family history of breast cancer—also contribute to individual risk but were not fully captured in this population-level analysis [41,51]. A healthy diet, abstaining from smoking, controlling body weight, and participating in regular exercise can reduce the number of risk factors linked to breast cancer. To successfully address these issues, initiatives supporting healthy lifestyles are essential, ranging from early education to government-sponsored advertising efforts.

## Projections for the next decade

Our analysis of the ten-year BC prediction shows a disturbing trend: the number of new cases among females is likely to increase during the next decade. Among the studied countries, Saudi Arabia is expected to experience the greatest rise, with nearly 20% increase in incidents rate comparing to than in the previous decade, reaching an estimated 9,441 additional cases. This finding was consistent with the worldwide potential rise. It was predicted that by 2040, both the incidence and death rates of breast cancer will have increased dramatically. The estimate predicts a whopping 40% growth in new cases each year, totaling almost 3 million cases worldwide. Furthermore, breast cancer-related mortality is predicted to increase by 50%, resulting in about 1 million deaths each year [39]. Demographic changes such as an aging population and changes in reproductive methods may have an influence on the reported patterns. As populations age and women postpone or opt to have fewer children, the risk of breast cancer increases. It was recommended that increasing females' awareness about breast cancer and how to avoid risk factors, along with early detection of breast cancer, can decrease the death rate from this disease [39,52,53]. The predicted increase differed between GCC countries for both females and men, with Saudi Arabia predicting the highest number of new BC cases among females and Bahrain among males. It is critical to extensively investigate these disparities to understand the different BC risk factors across the studied countries, such as lifestyle, healthcare systems, and government protection strategies [54,55]. Saudi Arabia and Bahrain must prioritize tackling this major health risk.

## Policy implications and recommendations

By including DALYs alongside incidence and mortality, this study sheds light on the full impact of breast cancer beyond just the number of cases and deaths. Our results show that DALY trends closely follow mortality patterns, underscoring the critical need for early detection and better cancer care. These findings provide valuable guidance for policymakers

to prioritize resources and develop health programs that not only aim to improve survival but also enhance quality of life. To tackle these challenges, several key actions are needed. First and foremost, public awareness campaigns stressing breast cancer risk factors and early detection should be combined with the implementation of comprehensive screening programs, which should specifically target high-risk populations. Secondly, in order to provide prompt access to diagnostic and treatment services, investments in healthcare infrastructure are essential. Thirdly, in addition to developing and implementing national policies that support efforts to prevent breast cancer, specific interventions for male breast cancer patients should also be created. To maximize effect, cooperation between government organizations, and healthcare professionals is also crucial. Finally, in order to lower obstacles to screening and treatment and increase access to care, socioeconomic inequities must be addressed.

## Strength and limitations

Utilizing the data from GBD 2019, this research offers a comprehensive evaluation of breast cancer burden and future projection. Notably, it represents the first attempt, to the best of our knowledge, to forecast the trajectory of age-standardized breast cancer incidence trends in GCC countries up to 2030. These findings provide valuable insight for policymakers, as they can inform future cancer control planning, resource allocation, and targeted prevention strategies. The identification of tobacco use and metabolic risks as leading contributors to breast cancer burden enables the development of region-specific health policies aimed at modifiable risk factors. Despite its strength, this study also bears certain limitations. Firstly, the predictive models did not account for specific significant risk factors such as family history in the estimation of future breast cancer incidence. Thus, the study is subject to confounding bias. Secondly, GBD data as a population-level data can lead to ecological fallacy, where associations observed at the population level are incorrectly attributed to individuals within that population. Thirdly, it's important to acknowledge that our study shares the general limitations associated with GBD research, including factors like the availability and quality of primary data.

Furthermore, estimates from the Global Burden of Disease (GBD) study rely on statistical models that help fill gaps in data, especially in countries where health information systems are still developing. In the Gulf Cooperation Council (GCC) countries, cancer registry systems differ in terms of coverage, completeness, and data quality. Challenges such as underreporting, inconsistent data collection methods, and delays in reporting can affect the accuracy of local cancer data. These issues may influence the reliability of GBD estimates, possibly leading to under- or overestimation of cancer burden in the region. Therefore, it is important to interpret these estimates with caution. Strengthening cancer registration systems and improving data collection practices across GCC countries will be essential for producing more accurate and useful public health information in the future.

## Supporting information

**S1 Table. Temporal trends of ASIR, ASDR, Prevalence, and DALY's by sex.**
(DOCX)

## Author contributions

**Conceptualization:** Majed M. Ramadan, Rawiah A. Alsiary.

**Data curation:** Majed M. Ramadan, Rawiah A. Alsiary.

**Investigation:** Majed M. Ramadan, Heba A Alkhatabi, Rawiah A. Alsiary.

**Methodology:** Majed M. Ramadan, Rawiah A. Alsiary.

**Software:** Majed M. Ramadan, Rawiah A. Alsiary.

**Supervision:** Majed M. Ramadan, Doaa Aboalola, Rawiah A. Alsiary.

**Validation:** Majed M. Ramadan, Rawiah A. Alsiary.

**Visualization:** Majed M. Ramadan, Heba A Alkhatabi, Rawiah A. Alsiary.

**Writing – original draft:** Majed M. Ramadan, Rawiah A. Alsiary.

**Writing – review & editing:** Majed M. Ramadan, Samkari Alaa, Rawiah A. Alsiary.

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
