## [Decision Letter · Decision Letter 0]

31 Mar 2025

PONE-D-25-05700Burden, and trends of breast cancer along with attributable risk factors in Gulf Cooperation Council Countries from 1990 to 2019 and its projections.PLOS ONE

Dear Dr. Alsiary,

Thank you for submitting your manuscript to PLOS ONE. After careful consideration, we feel that it has merit but does not fully meet PLOS ONE’s publication criteria as it currently stands. Therefore, we invite you to submit a revised version of the manuscript that addresses the points raised during the review process.

We look forward to receiving your revised manuscript.

Kind regards,

Ruo Wang

Academic Editor

PLOS ONE

Additional Editor Comments (if provided):

Reviewers' comments:

Reviewer's Responses to Questions

**Comments to the Author**

1. Is the manuscript technically sound, and do the data support the conclusions?

Reviewer #1: Yes

Reviewer #2: Yes

2. Has the statistical analysis been performed appropriately and rigorously? 

Reviewer #1: Yes

Reviewer #2: Yes

3. Have the authors made all data underlying the findings in their manuscript fully available?

Reviewer #1: Yes

Reviewer #2: Yes

4. Is the manuscript presented in an intelligible fashion and written in standard English?

Reviewer #1: Yes

Reviewer #2: Yes

5. Review Comments to the Author

Reviewer #1: Many thanks for preparation of this manuscript. I have read it fully. I think it is prepared well and in standard format that is defined for GBD-based manuscript.

However, please, dear authors are recommended to revise bellow points:

1. In line 216, "while the heist ASDR uptrend increase", Did dear authors mean "highest"?

2. I think the manuscript is well informative according to GBD database. However, the article suffers novelty. If dear authors can clarify the novel aspects of the article, it is appreciated to highlight it/them in a proper way.

3. Please, in "Strength and limitations" section, dear authors can described that how policy makers benefit from information presented in this article.

Again, many thanks with the best wishes.

Reviewer

Reviewer #2: This study systematically analyzed the epidemiological trends and risk factors of breast cancer in the GCC countries from 1990 to 2019 and predicted them to 2030. The study fills the gap in breast cancer data in the region and has reference value for public health strategy formulation. The data source is reliable and the method is generally reasonable. For the first time, GBD data were integrated to assess the burden and trend of breast cancer in the GCC countries, and the socioeconomic impact was analyzed in combination with HDI to provide a basis for resource allocation.

1. GBD data rely on estimation models, and the coverage and data quality of local cancer registration systems in GCC countries (such as whether there is underreporting) need to be clearly stated, and their potential impact on the results should be discussed.

2. The article mentions "excluding alcohol because of low consumption in GCC", but there is a lack of literature support (such as citing WHO or regional survey data) and quantitative evidence (such as the proportion of alcohol attributable in GBD).

3. The ASDR of Bahrain men increased significantly (39.13%), while it decreased in other countries, and the possible reasons need to be further explored (such as screening differences, changes in diagnostic criteria, or data heterogeneity).

4. The comparison between the increase in ASDR in Omani women and the decrease in Kuwait women needs to be further explained in combination with medical resources (such as the number of radiotherapy equipment), screening prevalence, or treatment differences.

In summary, the manuscript needs major revision.

6. PLOS authors have the option to publish the peer review history of their article (what does this mean? ). If published, this will include your full peer review and any attached files.

**Do you want your identity to be public for this peer review?** For information about this choice, including consent withdrawal, please see our Privacy Policy .

Reviewer #1: **Yes: ** Dr. Abdorrahim Absalan

Reviewer #2: No

---

## [Author Response · Author response to Decision Letter 1]

13 Apr 2025

Response to Reviewers' comments

Reviewer's Responses to Questions

Response: We sincerely thank both reviewers for their positive evaluations and constructive feedback. We are pleased to note that the manuscript was found to be technically sound, with appropriate statistical analyses and clear data availability. We also appreciate the acknowledgment that the manuscript is intelligibly written and meets the standards of written English.

1. Reviewer #1: Many thanks for preparation of this manuscript. I have read it fully. I think it is prepared well and in standard format that is defined for GBD-based manuscript.

However, please, dear authors are recommended to revise bellow points:

1. In line 216, "while the heist ASDR uptrend increase", Did dear authors mean "highest"?

Response: Thank you for pointing this out. Yes, you are correct — this was a typographical error. We intended to write "highest" instead of "heist." We have corrected this in the revised manuscript (page 9, line 263).

2. I think the manuscript is well informative according to GBD database. However, the article suffers novelty. If dear authors can clarify the novel aspects of the article, it is appreciated to highlight it/them in a proper way.

Response: We thank the reviewer for this valuable comment and the opportunity to clarify the novel contributions of our study. We have revised the manuscript to better highlight the originality of our work in both the Introduction (Page 3, line 136-140) and Discussion sections (first paragraph).

3. Please, in "Strength and limitations" section, dear authors can described that how policy makers benefit from information presented in this article.

Response: Thank you for your comment. The "Strength and limitations" section was modified accordingly.

Reviewer #2: This study systematically analyzed the epidemiological trends and risk factors of breast cancer in the GCC countries from 1990 to 2019 and predicted them to 2030. The study fills the gap in breast cancer data in the region and has reference value for public health strategy formulation. The data source is reliable and the method is generally reasonable. For the first time, GBD data were integrated to assess the burden and trend of breast cancer in the GCC countries, and the socioeconomic impact was analyzed in combination with HDI to provide a basis for resource allocation.

Response: We sincerely thank the reviewer for their positive and encouraging feedback. We appreciate your recognition of the novelty and value of our work in integrating GBD data to analyze the burden and trends of breast cancer in the GCC region.

1. GBD data rely on estimation models, and the coverage and data quality of local cancer registration systems in GCC countries (such as whether there is underreporting) need to be clearly stated, and their potential impact on the results should be discussed.

Response: Thank you for your valuable comment. We fully agree that while the Global Burden of Disease (GBD) study provides a comprehensive and standardized framework for estimating disease burden across regions and time, it relies on modeling techniques that are sensitive to the availability and quality of local data sources. In response to your suggestion, we have now added a dedicated paragraph in the strength and the limitation section to enhance transparency around the data limitations and encourage further investments in strengthening cancer surveillance systems in the region (pages 19-20, line 482–492).

2. The article mentions "excluding alcohol because of low consumption in GCC", but there is a lack of literature support (such as citing WHO or regional survey data) and quantitative evidence (such as the proportion of alcohol attributable in GBD).

Response: Thank you for this insightful observation. In response, we have revised the manuscript to include additional references supporting the exclusion of alcohol as a major risk factor in the context of breast cancer in the GCC region.

Please refer to the revised sections on pages 4, line 161 (ref. 9-11).

3. The ASDR of Bahrain men increased significantly (39.13%), while it decreased in other countries, and the possible reasons need to be further explored (such as screening differences, changes in diagnostic criteria, or data heterogeneity).

Response: Thank you for your comment. We agree. The necessary changes were implemented accordingly.

Please refer to the revised sections on pages 15, line 389-394.

4. The comparison between the increase in ASDR in Omani women and the decrease in Kuwait women needs to be further explained in combination with medical resources (such as the number of radiotherapy equipment), screening prevalence, or treatment differences.

Response: Thank you for your comment. We agree. The necessary changes were implemented accordingly.

Please refer to the revised sections on pages 14&15, line 367-378.

---

## [Decision Letter · Decision Letter 1]

25 Jun 2025

PONE-D-25-05700R1Burden, and trends of breast cancer along with attributable risk factors in Gulf Cooperation Council Countries from 1990 to 2019 and its projections.PLOS ONE

Dear Dr. Alsiary,

Thank you for submitting your manuscript to PLOS ONE. After careful consideration, we feel that it has merit but does not fully meet PLOS ONE’s publication criteria as it currently stands. Therefore, we invite you to submit a revised version of the manuscript that addresses the points raised during the review process.

We look forward to receiving your revised manuscript.

Kind regards,

Ruo Wang

Academic Editor

PLOS ONE

Journal Requirements:

Reviewers' comments:

Reviewer's Responses to Questions

**Comments to the Author**

1. If the authors have adequately addressed your comments raised in a previous round of review and you feel that this manuscript is now acceptable for publication, you may indicate that here to bypass the “Comments to the Author” section, enter your conflict of interest statement in the “Confidential to Editor” section, and submit your "Accept" recommendation.

Reviewer #3: All comments have been addressed

Reviewer #4: (No Response)

Reviewer #5: All comments have been addressed

Reviewer #6: (No Response)

Reviewer #7: (No Response)

2. Is the manuscript technically sound, and do the data support the conclusions?

Reviewer #3: Yes

Reviewer #4: Yes

Reviewer #5: Yes

Reviewer #6: (No Response)

Reviewer #7: Partly

3. Has the statistical analysis been performed appropriately and rigorously? 

Reviewer #3: Yes

Reviewer #4: I Don't Know

Reviewer #5: Yes

Reviewer #6: (No Response)

Reviewer #7: Yes

4. Have the authors made all data underlying the findings in their manuscript fully available?

Reviewer #3: Yes

Reviewer #4: Yes

Reviewer #5: Yes

Reviewer #6: (No Response)

Reviewer #7: Yes

5. Is the manuscript presented in an intelligible fashion and written in standard English?

Reviewer #3: Yes

Reviewer #4: Yes

Reviewer #5: Yes

Reviewer #6: (No Response)

Reviewer #7: Yes

6. Review Comments to the Author

Reviewer #3: Thank you for this thorough revision. The revised manuscript convincingly addresses nearly all of the substantive issues raised during the previous round of review: the new paragraph in Strengths & Limitations candidly discusses data-quality caveats, the Methods section now provides a clearer rationale for excluding alcohol as a risk factor, and the expanded Discussion offers plausible explanations for the country-specific trends we highlighted.

Reviewer #4: Thank you for your efforts for this study. The study elucidates some substantial epidemiological and attributable risk rates of breast cancer in GCC countries.

The following corrections and revisions are recommended for implementation.

1) In abstract, the measurements employed in this study (ASR, EAPC, etc.) should be placed in the 'material and methods' section.

2) Line 134. It provides an explanation of the ASR measurement rather than the EAPC, and this should be corrected.

3) Line 120. What is the cut-off points of high meat diet? As stated others (BMI, plasma glucose etc), please provide the necessary information also for this.

4) Line 124. What is the reference level for tobacco use? It is recommended that both 'dietary risk' and 'tobacco use' be assigned adequate cut-off points.

5) Line 290. The word 'counties' should be replaced with 'countries'.

6) References of figures should be placed as a footnote under the Figure. The source of these references is not clear. It is advisable to ascertain whether the necessary permissions have been obtained.

Reviewer #5: This study represents comprehensive report to breast cancer in GCC countries with valuable insights; however, it lacks unmeasured confounders as genetics or environmental exposure that mostly contributes to disease progression. Overall, it contributes to baseline foundation. The research report is well-written and addressed but it still needs revision for all grammar and punctuation issues.

Reviewer #6: This manuscript offers a comprehensive analysis of breast cancer trends in GCC countries from 1990 to 2019, with projections to 2030. Using Global Burden of Disease data, the study highlights rising incidence rates, particularly in Saudi Arabia, and links them to HDI, metabolic risks, and tobacco use. The findings emphasize the need for targeted awareness, early detection, and healthcare improvements across the region.

The revised version meets the criteria of publication.

Reviewer #7: This paper is quite ambitious in comprehensiveness, analyzing the last 20 years and 10-year projections across multiple trends in breast cancer. Reporting these results alone is a great contribution to the literature as well as helping individuals, communities and policymakers in these countries understand recent trends as well as projected trends.

With these primary contributions in mind, there are aspects of the paper that are underdeveloped, distract or overreach in terms of author claims. The paper would be stronger by removing these aspects (e.g. DALY, HDI) and focus simply on reporting the past trends and future projections.

7. PLOS authors have the option to publish the peer review history of their article (what does this mean? ). If published, this will include your full peer review and any attached files.

**Do you want your identity to be public for this peer review?** For information about this choice, including consent withdrawal, please see our Privacy Policy .

Reviewer #3: No

Reviewer #4: No

Reviewer #5: No

Reviewer #6: No

Reviewer #7: No

---

## [Author Response · Author response to Decision Letter 2]

13 Jul 2025

Reviewers' comments

1. General Comments

This paper conducts a thorough analysis of trends in breast cancer incidence, prevalence and death rates, for both women and men across 6 GCC countries for the period 1990-2019. Further, the authors make annual projections age-standardized incidence rates for the next decade (2020-2030) for the same countries. This paper is quite ambitious in comprehensiveness, analyzing the last 20 years and 10-year projections across multiple trends in breast cancer. Reporting these results alone is a great contribution to the literature as well as helping individuals, communities and policymakers in these countries understand recent trends as well as projected trends.

With these primary contributions in mind, I offer the following comments. In particular I address aspects of the paper that are underdeveloped, distract or overreach in terms of author claims.

Response: We sincerely thank the reviewer for their positive assessment of our work and for recognizing the contribution of our study to both the academic literature and policymaking. In the revised manuscript, we have carefully addressed each of the reviewer’s specific comments to enhance the clarity, balance, and scientific rigor of our analysis and interpretations. Detailed responses are provided below.

2. Comment on DALYs analysis throughout paper

Early on in the paper, the authors focus on breast cancer incidence, prevalence and death rates. Not until the Methodology section is the measure of the DALY introduced. (In a similar vein, there are multiple references to prevalence estimates early in the paper though very little is reported on prevalence results such as Fig 2.) The inclusion of DALYs (e.g. Table 1 of Results), seems more a distraction from the main contribution of the incidence and death rates, past and projected. If the DALY burden rates remain in the paper, include explicit motivation in the introduction. Further, guidance should be provided to the reader on these results in the discussion section of the paper.

Response: Thank you for this valuable observation. We agree that the DALY analysis was not clearly motivated in the original version. In response, we have (1) added a brief justification for including DALYs in the Introduction, emphasizing their relevance to assessing overall disease burden, and (2) expanded the Discussion to provide clearer interpretation and policy relevance of the DALY findings. We have also reviewed the placement of DALY results to ensure they complement rather than distract from the core focus on incidence and mortality.

3. Comment on analysis of results for men

While breast cancer affects both men and women, the magnitude of the incidence, death and DALYs rates for women is significantly higher than for men (Table 1). As such, reporting the average effects is not meaningful. If results for men remain in this paper (see next paragraph), Table 1 might better consist of two panels – reporting for women only and then men only, across the GCC countries.

Factors that may be associated with breast cancer diagnoses for men and women may be quite different. Further, the approach that health care providers may use to support screening and health care service and support is likely to be quite different for men and women. All of this to say that it might be worthwhile to further explore these trends for men only in a separate paper.

Reponses: We appreciate the reviewer’s thoughtful comment regarding the presentation of male breast cancer data. We agree that the burden of breast cancer is substantially higher among women compared to men. However, the primary purpose of this manuscript is to provide comprehensive and inclusive data to inform policymakers across GCC countries for future planning and resource allocation. The inclusion of data on men, despite their smaller burden, is necessary to fully characterize the epidemiology of breast cancer in the region and to ensure that male breast cancer cases are not overlooked in national cancer control strategies. In line with this purpose, we have followed the approach adopted by the Global Burden of Disease (GBD) Study in its Lancet publications in presenting gender, which included men to provide a complete assessment of disease burden across populations. This approach was confirmed with our first co-author, a senior GBD collaborator, who emphasized the importance of including male data for comparability and policy relevance. Additionally, we note that we have reached the maximum allowable limit of tables for this journal. Therefore, splitting Table 1 into two separate panels is not feasible within the current manuscript format.

We hope this clarifies our rationale for retaining the current table structure and presentation approach. We appreciate the reviewer’s suggestion to explore male breast cancer trends in a separate analysis and will consider this for future dedicated publications.

We hope this clarifies our rationale for retaining the current table structure and presentation approach. We appreciate the reviewer’s suggestion to explore male breast cancer trends in a separate analysis and will consider this for future dedicated publications.

4. Comment on attributable risk factors

At several points throughout the paper – e.g., introduction (lines 53-57), references to DALYs in methodology and results sections & discussion – the statements regarding attributable risk factors uses language that makes much more of a causal claim than is suggested by the evidence. The authors make a stronger case if they reported results as associations or correlations with socioeconomic and environmental factors. If there is significant evidence regarding the causal relationship between one or two of the environmental factors and breast cancer diagnosis, I recommend a more extensive discussion of those factors directly, in the same way they outline exclusion of alcohol consumption from their analysis.

Response: We thank the reviewer for their valuable comments. In response, we have thoroughly revised the manuscript to replace language-suggesting causality with terminology that more accurately reflects associations or correlations between breast cancer burden and socioeconomic and environmental factors. Where strong evidence of causality exists—such as for metabolic factors and tobacco use—we have expanded the discussion accordingly, consistent with our approach to alcohol consumption exclusion. Additionally, we have introduced a new subheading in the Discussion titled “Metabolic and Lifestyle Factors in Breast Cancer: Evidence and Insights.”

These revisions enhance the clarity and scientific rigor of our findings while ensuring alignment with the current evidence.

5. Comment on ASIR & HDI

Similar to my previous comment, I am not certain what to make of the statements regarding the relationship between ASIR and the Human Development Index (HDI). At a minimum, more explanation is required to help the reader understand your hypothesis about the association of the ASIR and HDI. How should the reader think about the correlation between these two measures, particularly given that the results (starting line 197) show significant positive correlation for some countries and negative for others. The discussion section (starting line 311) provides speculation and hypothesizing for those countries with the positive association only.

Addressing this may be better suited as a question for future research.

Response: Thank you for your helpful observation. We acknowledge that our previous wording may have caused confusion. As shown in our results (line 197), there is a consistent, statistically significant positive correlation between ASIR and HDI across all countries, with the strongest observed in Saudi Arabia (r=0.99; p <.0001). We have revised the Discussion section to:

• Clearly restate that ASIR is positively correlated with HDI in all GCC countries.

• Clarify that there are fluctuations in correlations between ASDR, DALYs, and HDI across countries

• Expand our interpretation to include potential explanations for our finding

These changes aim to improve accuracy, clarity, and completeness in interpreting the HDI associations.

6. Comment on organization of the Discussion section

The discussion section is lengthy and could benefit from the addition of subheadings. This will help guide the reader through your wide-ranging and comprehensive analysis – across gender, time, as well as countries.

As an example, one might break down this section as follows:

Discussion - ASIR for women (line228)

Discussion - ASIR men (line 283)

Discussion - ASIR & HDI (line 311)

Discussion - 10yr Projections (line 342)

Policy Recommendations (line 360)

Response: Thank you for this helpful suggestion. We agree that subheadings will enhance the reader’s ability to follow and interpret the results and implications of our study. Thank you again for this valuable recommendation. We have revised the discussion section by adding the following subheadings at the suggested points:

• ASIR Trends in Women

• ASDR Trends in Men

• Association Between ASIR and HDI

• Projections for the Next Decade

• Policy Implications and Recommendations

---

## [Decision Letter · Decision Letter 2]

13 Aug 2025

Burden, and trends of breast cancer along with attributable risk factors in Gulf Cooperation Council Countries from 1990 to 2019 and its projections.

PONE-D-25-05700R2

Dear Dr. Alsiary,

We’re pleased to inform you that your manuscript has been judged scientifically suitable for publication and will be formally accepted for publication once it meets all outstanding technical requirements.

Kind regards,

Ruo Wang

Academic Editor

PLOS ONE

Additional Editor Comments (optional):

Reviewers' comments:

Reviewer's Responses to Questions

**Comments to the Author**

1. If the authors have adequately addressed your comments raised in a previous round of review and you feel that this manuscript is now acceptable for publication, you may indicate that here to bypass the “Comments to the Author” section, enter your conflict of interest statement in the “Confidential to Editor” section, and submit your "Accept" recommendation.

Reviewer #5: All comments have been addressed

Reviewer #6: (No Response)

Reviewer #7: (No Response)

2. Is the manuscript technically sound, and do the data support the conclusions?

Reviewer #5: Yes

Reviewer #6: (No Response)

Reviewer #7: Yes

3. Has the statistical analysis been performed appropriately and rigorously? 

Reviewer #5: Yes

Reviewer #6: (No Response)

Reviewer #7: Yes

4. Have the authors made all data underlying the findings in their manuscript fully available?

Reviewer #5: Yes

Reviewer #6: (No Response)

Reviewer #7: Yes

5. Is the manuscript presented in an intelligible fashion and written in standard English?

Reviewer #5: Yes

Reviewer #6: (No Response)

Reviewer #7: Yes

6. Review Comments to the Author

Reviewer #5: Manuscript appears suitable for acceptance. The study provides a predictive insight in risk factors associated to breast cancer in gulf countries. Manuscript is well-addressed.

Reviewer #6: The revised version of the paper has sufficiently addressed all the concerns I had. Thus, I would recommend accepting it.

Reviewer #7: Thank you for your responses to my comments on the previous version and for the opportunity to read the revised paper. Reporting these trends is an important contribution to the literature. Please see my review for my full reply and minor edits for clarification.

7. PLOS authors have the option to publish the peer review history of their article (what does this mean? ). If published, this will include your full peer review and any attached files.

**Do you want your identity to be public for this peer review?** For information about this choice, including consent withdrawal, please see our Privacy Policy .

Reviewer #5: No

Reviewer #6: No

Reviewer #7: No

---

## [Editor Report · Acceptance letter]

PONE-D-25-05700R2

PLOS ONE

Dear Dr. Alsiary,

I'm pleased to inform you that your manuscript has been deemed suitable for publication in PLOS ONE. Congratulations! Your manuscript is now being handed over to our production team.

Kind regards,

on behalf of

Dr. Ruo Wang

Academic Editor

PLOS ONE